# Differences in Urban Morphology between 77 Cities in China and Europe

**Fengxiang Guo** [1] [iD], **Uwe Schlink** [1,*] [iD], **Wanben Wu** [1,2] and **Abdelrhman Mohamdeen** [1]

[1] Department of Urban and Environmental Sociology, UFZ—Helmholtz Centre for Environmental Research, 04318 Leipzig, Germany

[2] Ministry of Education Key Laboratory for Biodiversity Science and Ecological Engineering, National Observations and Research Station for Wetland Ecosystems of the Yangtze Estuary, Shanghai Institute of EcoChongming (SIEC), Fudan University, Shanghai 200433, China

[*] Correspondence: uwe.schlink@ufz.de; Tel.: +49-341-235-1554

**Abstract:** Urban morphology refers to the physical form of a city that is constantly transformed and updated in the process of urbanization. A valuable source of data on 'built forms' is modern remote sensing technology, which provides a variety of products on building footprints and heights at national, continental, and global levels. A large-scale comparison of urban morphologies is important for assessing urban development as well as its influence on urban ecology; however, this has not been well documented so far. This study includes 41 cities in China and 36 in Europe with various city sizes, population densities, and climate features. We applied 3D landscape metrics and principal component analysis (PCA) to compare the spatial aspects of the urban morphology of these cities. We found: (1) measurements of the building height, surface fluctuation, and texture directionality of urban building layouts in China are higher than those of European cities, while the latter are high-density and compact built landscapes; (2) a significant clustering phenomenon for Chinese and European cities revealed by PCA, with the former showing a much more aggregated pattern, indicating a relatively uniform morphology of urban buildings in China; (3) distinctions between cities in China and Europe are suggested by the first principal component, to which building height, surface fluctuation, building complexity, and spatial distance among buildings contribute significantly; and (4) the second principal component (mainly represented by maximum building height, surface area, volume, and shape metrics) can separate large metropolitan cities and provincial capitals from cities with lower urban population, smaller size, and slower economic development. Our results demonstrate the potential of 3D landscape metrics for measuring urban morphology. Together with a temporal analysis, these metrics are useful for quantifying how urban morphology varies in space and time on a large scale, as well as evaluating the process of urbanization.

**Keywords:** urban morphology; 3D landscape metrics; Chinese and European buildings; principal component analysis; cluster

## 1. Introduction

Climate change is a pressing issue today and makes cities increasingly suffer from extreme weather events, such as flooding and heat episodes. The adaptation of cities to long-term environmental changes demands a transformation of their morphology. Interdisciplinary studies of urban morphology focus on the spatial analysis of urban structures, land use, street patterns, buildings, and open spaces [1,2]. Providing a complex reflection of human–nature interactions in cities, urban morphology involves a wide range of landscape features, characteristics of urban architecture, and ecological parameters, including air quality, ventilation, traffic noise, and accessibility [3–6]. Of particular interest is the formation of the urban thermal landscape, which involves the urban heat island (UHI) phenomenon [7–9]. In the UHI, the heat balance of cities is considerably modified compared with their surroundings, and this depends on urban structures as well as on the climate.

Urban morphology refers to the physical form of a town or city. It is defined by buildings and their related open spaces, plots or lots, and streets [1]. These elements undergo continuous transformation and replacement—these processes can be used for the adaptation to a changing environment. The morphological elements are considered at specified levels of spatial resolution and time periods. To facilitate studies exploring associations between urban structures and the urban thermal environment, this paper aims to suggest a set of characteristics of the urban morphology that are useful for heat studies. While specific studies of associations have been made previously [8,9], here, we focus on the definition of morphological characteristics that can be derived from remote sensing data. For that purpose, we follow a spatial analytical approach [10], analyzing 'built forms' derived from satellite images at high resolution. Various remote sensing data sources have been applied to extract building information, such as light detection and ranging (LiDAR) data [11], high-resolution remote sensing imagery [12], synthetic aperture radar (SAR) images [13,14], and aerial imagery [15]. However, large-scale studies of urban morphology are not well documented due to discontinuous data collection processes and high costs.

The restricted availability of methods to describe and analyze building forms in 3D space is another limitation for a large-scale comparison. Common metrics for measuring urban building characteristics mainly include building height, building density, and the spatial proportion of various building classes [8,15]. These metrics focus on composition characteristics and can describe the fluctuation of building surfaces to a certain extent; however, they ignore configuration characteristics (e.g., building compactness, texture, and diversity). Although some studies chose the sky view factor as an openness indicator by measuring sky visibility, in addition to the fractal dimension metric as a complexity indicator in a built landscape [8,16–18], building details are still insufficiently involved. A weakness in these studies is that building texture is rarely considered, which refers to the regularity and directionality of the spatial arrangement of urban buildings and can directly affect urban ventilation. Benefitting from scientific planning, modern building layouts are usually much more regular and disordered in comparison to traditional and historical building layouts [8]. Therefore, systematic and efficient indicators for measuring urban morphology are particularly important.

Landscape metrics for pattern recognition are widely applied to measure the spatial heterogeneity and fragmentation of natural landscapes [18–20]. These metrics were designed from two functional aspects: (i) landscape composition, i.e., the relative proportion of LULC, and (ii) spatial configuration, i.e., the spatial arrangement of patch types [21]. Traditional landscape metrics are capable of 2D raster data; however, building objects actually refer to 2.5D or 3D representation [8]. To deal with this insufficiency, Kedron et al. (2019) and Guo et al. (2021) proposed a set of 3D landscape metrics for building objects that aim to systematically analyze the built landscape, and they tested their effectiveness in various cities across space and time [8,22]. Despite some interesting metrics (e.g., building spacing, building shape, and richness metrics), surface metrics, particularly texture and fluctuation parameters, are ignored. Building texture, particularly directionality, is vital to the urban ventilation effect, which has a direct influence on urban heat [8]. Wu et al. (2017) demonstrated how surface metrics can be computed in 3D space; however, they did not test them for urban built landscapes [23]. According to their work, the texture direction of building layouts can be calculated from the spatial autocorrelation spectrum using buildings' footprint and height data. In this study, we focus on building characteristics that reflect the differences in urban morphology between Chinese and European cities, aiming to find significant differences between these cities and to explain which building characteristics are causing such differences.

## 2. Materials and Methods

### 2.1. Study Region

In this study, we selected 77 cities (including 41 cities in China and 36 in Europe) based on the criteria of city size, economic level, urban population, and climate features

(Figure 1). For China, most of these cities are metropolitan or provincial capitals with a population density over 1000/km², gross domestic product (GDP) over CNY 14 billion, and a whole-city size larger than 1900 km² (Table A1 in Appendix A). The selected Chinese cities are mainly distributed in five climate zones, including severe cold, cold, hot-summer–cold-winter, hot-summer–warm-winter, and temperate regions. In each country or region in Europe, a typical city was selected, usually the country's capital, to ensure the diversity and comparability of urban morphology. Europe has three main climate zones—marine west coast, humid continental, and the Mediterranean. Five additional climate zones appear in small areas of Europe—subarctic, tundra, highland, steppe, and humid subtropical. Selected European cities cover almost all these climate zones.

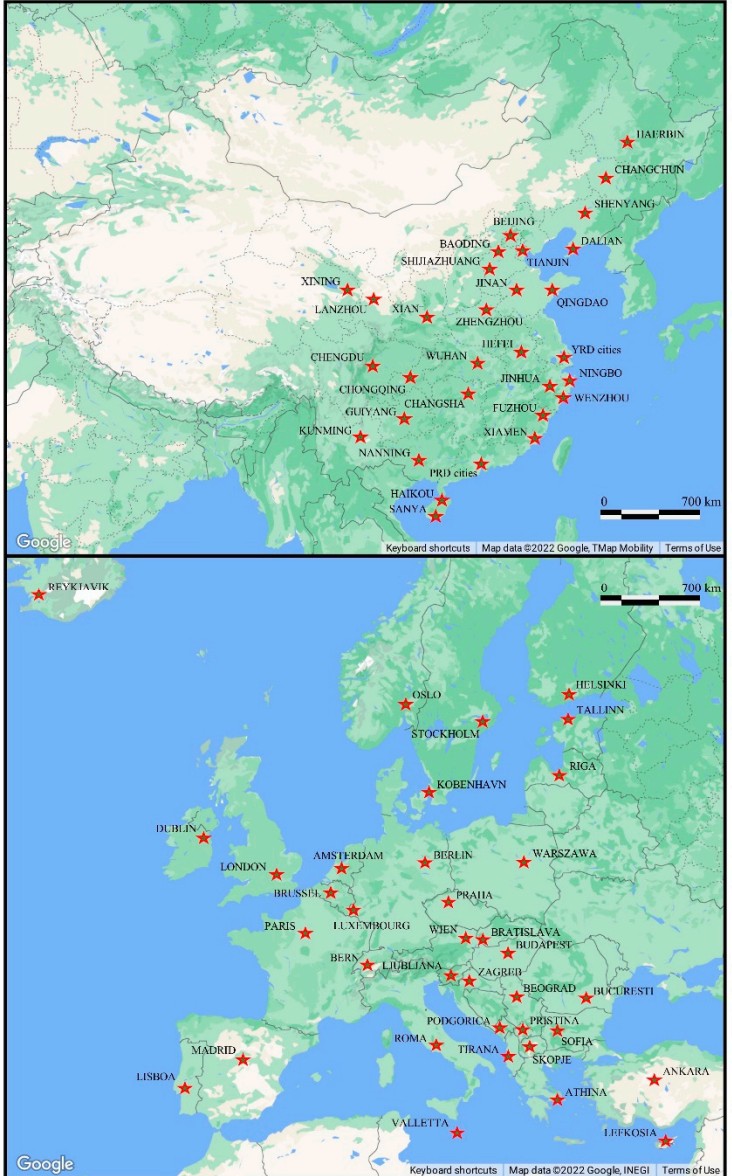

**Figure 1.** Locations of selected cities in China and Europe. In China, YRD (Yangtze River Delta) cities include Shanghai, Nanjing, Jiaxing, Shaoxing, Hangzhou, Suzhou, and Wuxi. PRD (Pearl River Delta) cities include Guangzhou, Shenzhen, Dongguan, Huizhou, Foshan, and Zhuhai.

*2.2. Data*

The data applied in this study mainly refer to the building's footprint and height. For China, data is downloaded from the Resource and Environment Science and Data Center,

Chinese Academy of Science, which originates from Baidu Map services. Original data type is vector file using UTM projection with an overall accuracy of the building height product of 86.78% [20]. For Europe, the building data comes from the European Union's Earth observation program Copernicus with 10 m spatial resolution raster using the Slovak Basic Database for the Geographic Information System (ZBGIS). The building data was collected in 2012, and relative accuracy descriptions of building heights are attached with the raster data. In addition, Szatmári et al. [24] verified the accuracy of building heights for the Europe dataset. Four different tests showed that an error larger than 6 m fell in the interval 17–30%. This accuracy is sufficient for large-scale environmental analyses [24].

*2.3. Methods*

2.3.1. Urban Building Metrics

This study used previously proposed 3D landscape metrics [8,23] to measure spatial heterogeneity of urban morphology. These metrics are designed based on the patch mosaic model and the digital terrain model, which are applicable to discrete data. Once categorized, boundaries of patches were defined [19,22,25]. An urban built landscape can be seen as a mixture of multiple buildings within a certain area in 3D space [8,22].

To calculate the metrics, firstly we divided all buildings into several classes based on their height. Considering a lower building height in Europe than in China, this study applied the European classification standard and divided built landscapes into 5 classes: low buildings (below 6 m), sub-low-rise buildings (6 m–10 m), middle-rise buildings (10 m–20 m), sub-high-rise buildings (20 m–60 m), and high-rise buildings (over 60 m). Four heterogeneity levels of cells, patches, classes, and landscapes should be defined: 'cell' refers to building pixel in classified raster data, 'patch' refers to a closed building region where the internal building pixels have similar height attributes but are significantly different from the outside pixels, and 'class' refers to the mixture of different building patches with the same or similar height.

The selected landscape metrics include composition and configuration metrics, aiming to characterize the diversity, complexity, compactness, and spatial arrangement regularity of urban buildings (Table 1). The composition metrics are further divided into 2D and 3D metrics depending on whether 3D vertical landscape elements were taken as main variable for calculation or not. In addition, this study implemented the relevant software LPA3D in a MATLAB library, which is important for a straightforward application of these metrics.

**Table 1.** Selected 3D landscape metrics for measuring urban morphology. For detailed calculation steps, please see literature [8,23].

| Metrics | Abb. | Type | Measure of the . . . |
|---|---|---|---|
| Patch density | PD | Composition-2D | spatial heterogeneity and evenness of urban building patterns. |
| Euclidean nearest-neighbor mean distance | ENN | Composition-2D | isolation degree of each building's class, and can be taken as indicator for measuring road width. |
| Percentage of patch type | PLAND | Composition-2D | proportion of each building's class in the urban building pattern. |
| Edge density | ED | Composition-3D | boundary density of urban buildings. |
| Mean building height | BH | Composition-3D | mean height of urban buildings. |
| Maximum building height | BHMAX | Composition-3D | highest building height. |

**Table 1.** *Cont.*

| Metrics | Abb. | Type | Measure of the . . . |
|---|---|---|---|
| Surface area | SA | Composition-3D | surface fluctuation compared with plane area. |
| Mean volume index | VOL | Composition-3D | mean volume of urban buildings. |
| Standard deviation of height | SQ | Composition-3D | undulation of the urban building's surface. |
| Surface skewness | SSK | Composition-3D | SSK > 0, which represents more building height, while SSK < 0 represents less building height than an average plane. |
| Surface kurtosis | SKU | Composition-3D | spatial distribution of extreme building height conditions. |
| Building surface slope | SSL | Composition-3D | integral slope of building surface, which is the sum of surface fluctuation at adjacent building pixels. |
| Texture direction aspect ratio | STR | Configuration | building surface texture direction. STR approaches 1, meaning building pattern has no dominant orientation; STR approaches 0, meaning building pattern has dominant orientation (Figure 2). |
| Building shade metrics | BSM | Configuration | effect of buildings forming ventilation paths, defined by the ratio between building height and spacing (ENN). |
| Building object to building patch number ratio | BN2PN | Configuration | complexity and fragmentation of buildings. Each individual building object might be divided into several patches due to height differences. |
| Largest patch index | LPI | Configuration | largest space occupation of single building. |
| Landscape shape index | LSI | Configuration | deviation between patch shape and regular circle or square with same area. |
| Cohesion index | COI | Configuration | connectivity and aggregation of the urban building pattern. |
| Effective mesh size | MESH | Configuration | fragmentation and aggregation of urban buildings landscape. |
| Shannon's diversity index | SHDI | Configuration | diversity of urban buildings landscape. |

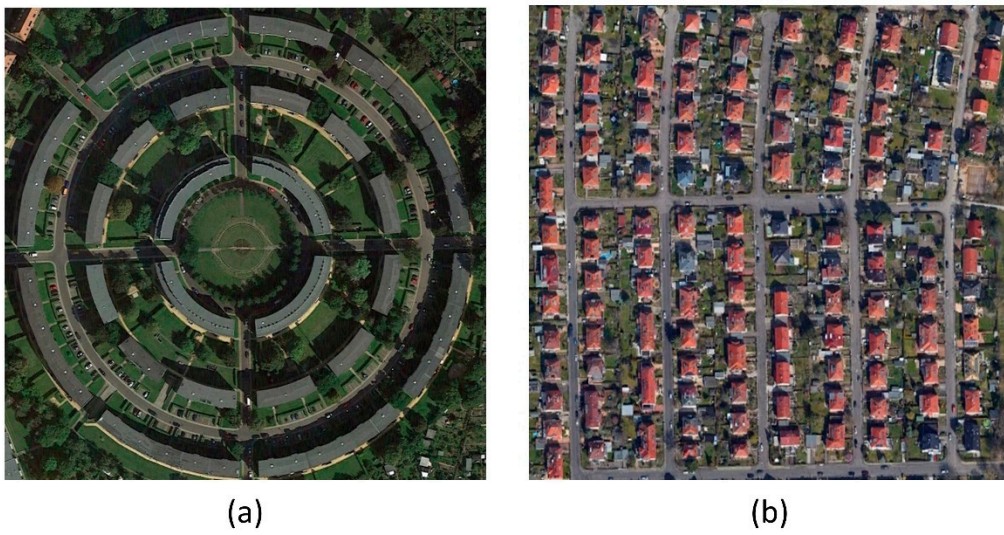

(a)       (b)

**Figure 2.** Texture features of different building layouts (image source: Google Earth); (**a**) displays significant isotropy, while (**b**) displays significant anisotropy.

2.3.2. Principal Component Analysis

Principal component analysis (PCA) is a dimensionality reduction method that can retain trends and patterns [26,27]. PCA is an orthogonal linear transformation of multiple variables or high-dimensional data into a new coordinate system, sometimes only using few important components and neglecting the rest. These components are independent and uncorrelated [26]. PCA starts from the matrix $F(c)$, which contains building features in the rows for the cities arranged in the columns.

$$F(c) = \begin{bmatrix} f_1(c_1) & f_1(c_2) & \cdots & f_1(c_n) \\ f_2(c_1) & f_2(c_2) & \cdots & f_2(c_n) \\ \vdots & \vdots & \ddots & \vdots \\ f_m(c_1) & f_m(c_2) & \cdots & f_m(c_n) \end{bmatrix} \tag{1}$$

where $f_i$ $(i = 1, 2, \cdots, m)$ represents the $i$-th building feature (out of $m = 24$) and $c_i$ $(i = 1, 2, \cdots, n)$ represents the $i$-th city in Europe or China (out of $n = 77$ cities).

Computing the mean for every dimension of the whole dataset using Equation (2),

$$\overline{f_i} = \frac{1}{m} \sum_{j=1}^{n} f_i(c_j) \tag{2}$$

We normalized the data matrix as follows:

$$F'(c) = \begin{bmatrix} f_1(c_1) - \overline{f_1} & f_1(c_2) - \overline{f_2} & \cdots & f_1(c_n) - \overline{f_n} \\ f_2(c_1) - \overline{f_1} & f_2(c_2) - \overline{f_2} & \cdots & f_2(c_n) - \overline{f_n} \\ \vdots & \vdots & \ddots & \vdots \\ f_m(c_1) - \overline{f_1} & f_m(c_2) - \overline{f_2} & \cdots & f_m(c_n) - \overline{f_n} \end{bmatrix} \tag{3}$$

We computed the covariance matrix of the whole dataset

$$COV = \frac{1}{m} F'^T F' \tag{4}$$

The last step is to compute eigenvectors $\psi_i$ and the corresponding eigenvalues $\lambda_i$ using Equation (5) [26]:

$$(COV - \lambda_i I)\psi_i = 0 \tag{5}$$

where $I$ represents the identity matrix ($m \times m$). After sorting the eigenvalues by decreasing eigenvalues ($\lambda_1 > \lambda_2 > \cdots > \lambda_m$), we choose $k$ eigenvectors with the largest eigenvalues to form $n \times k$ dimensional matrix $W$, from which we obtain the principal component score $S_{pca}$ and principal component coefficients $C_{pca}$. The centered data can be reconstructed by $S_{pca} \times C_{pca}'$. Usually, most of the variance is contained in the first few components, which retain important information with less noise. In this case, PCA reduces the degrees of freedom and removes noise, and the first few components are seen as sensitive features for detecting patterns [26–28].

## 3. Results

### 3.1. Different Metrics of City Groups in China and Europe

For a better comparison of the spatial heterogeneity of urban buildings in China and Europe, standardization of the calculated metrics was firstly conducted. Among the 2D composition metrics, European cities have higher building density (PD), narrower distance among building patches (ENNPA), and more low-rise and sub-low-rise buildings (LB and SLB), while the buildings in selected Chinese cities are much higher (SHB and HB) and the buildings spacings are wider (ENNPA). Among the 3D composition metrics, building height (BH), fluctuation of building surface (SQ), and building edge density (ED) in China significantly exceed those of Europe (Figure 3).

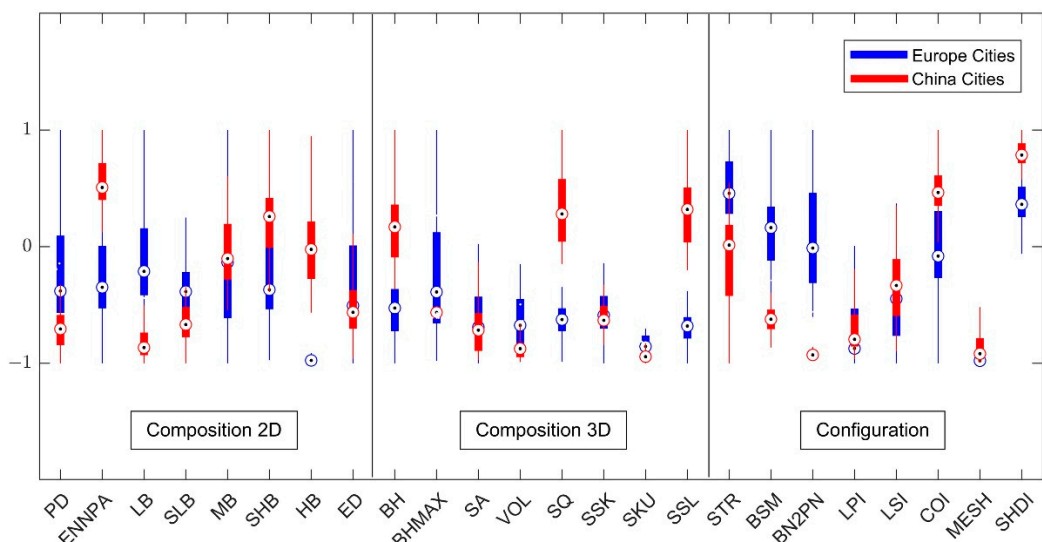

**Figure 3.** Results of 3D landscape metrics belonging to three groups. Red color represents Chinese cities, while blue color represents European cities.

Maximum building height (BHMAX), surface area (SA), and volume (VOL) did not show significant differences between the two regions. In spite of higher building height and wider building spacing in China, their ratio (BSM) was much lower than that in Europe, which indicated a stronger influence of building spacing. Together with building density, both reflected a much more compact building layout in European cities. For other configuration metrics, the spatial arrangement of urban buildings might be more isotropic in Europe, with higher STR values than those in China. A lower STR value means an anisotropic tendency in urban building layouts. In addition, the building diversity and building fragmentation levels in China also exceeds those of Europe. The differences may be related to the urbanization. China is witnessing giant changes in urban form. In most of the selected cities, historical buildings, traditional houses, and modern buildings are all included. Compared with history and traditional buildings, modern buildings are planned in a much more regular way.

### 3.2. More Uniform Urban Morphology in China Than in Europe

PCA results indicated a significant cluster phenomenon for Chinese and European cities. The former displayed a much more aggregated pattern than those in the latter through comparing the longest radius of circumscribed ellipse for these two regions (Figure 4a–c). This suggests that the urban morphology of Chinese cities might be relatively uniform compared with Europe cities, which might be related to urbanization level, urban population, and local architecture culture.

The first two components together explain 64.5% of variations, while Component 3 additionally explains 9.3% of the variance. Figure 4b,c indicate that Component 1 can distinguish well between cities in China and Europe. It is obvious that most of the European cities are characterized by a negative value of Component 1, while cities of China all have a positive value of Component 1 (Figure 4c). By ranking the relative contribution of building metrics to Component 1, we found that building height, spatial proportion of different building classes, building surface fluctuation, the complexity of buildings, and distance among buildings contributed more than others (Figure 5). The results in Section 3.1 demonstrate that Chinese cities have higher urban buildings, larger surface fluctuations, and wider building distances.

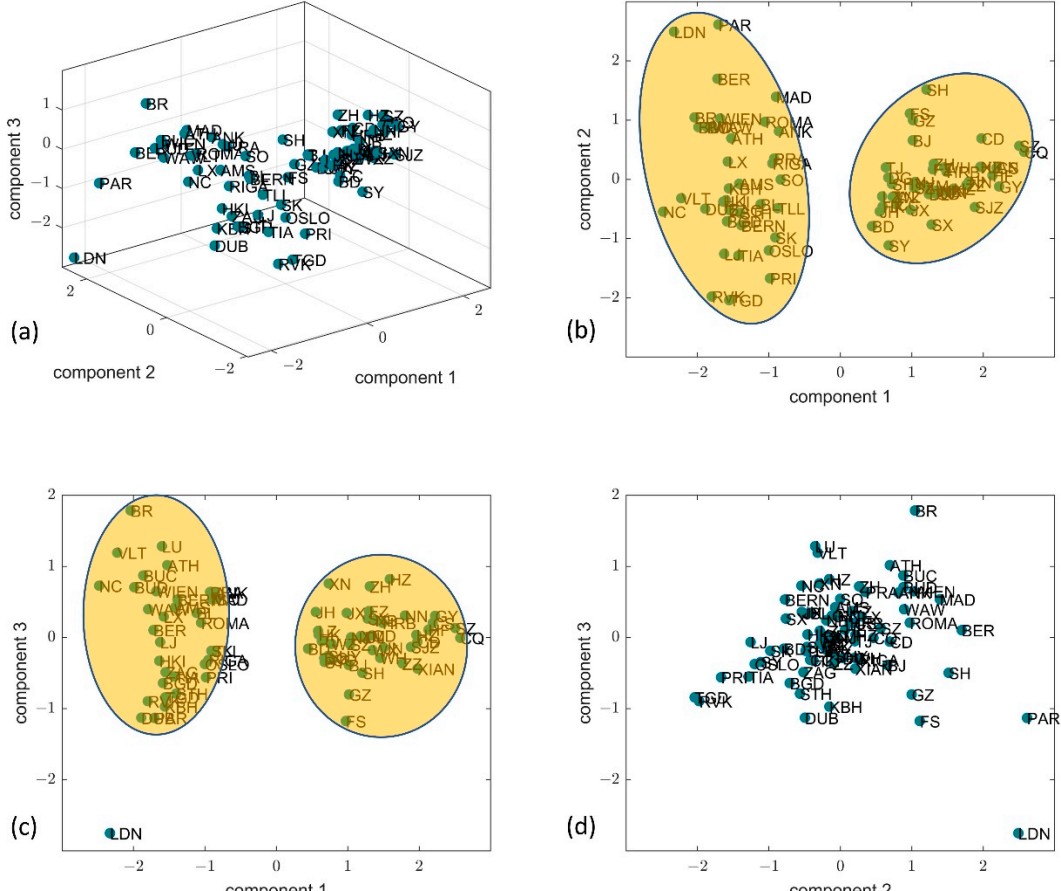

**Figure 4.** (**a**) Results of the first, second, and third principal components of building metrics calculated for cities in China and Europe; (**b**) the first component versus second component; (**c**) the first component versus third component; (**d**) the second component versus third component.

Figure 4b also shows that the second component is sensitive to the degree of urban development and does not reflect the geographic region as the first component does. To be specific, no matter whether it is for Europe or China, higher Component 2 values indicate international metropolitan cities and province capitals. In Western Europe, major metropolitan cities, such as London, Paris, Madrid, and Berlin, had higher values of Component 2 than cities in Eastern and Northern Europe. The Northern European cities display a much lower level of Component 2, in particular Reykjavik, which has almost the lowest. Compared with European cities, the spatial distributions of Component 2 for Chinese cities are much more aggregated. Higher values of Component 2 were found in Shanghai, Beijing, Chengdu, and Shenzhen, with fast urbanization occurring over the past few decades. These cities kept a similar level with cities in Western Europe cities, except London and Paris, which show the dominant characteristics in Component 2 (e.g., shape metrics, surface area, volume, edge density, and cohesion metrics), and these cities may be similar, as is suggested by Figure 5. For other cities in China, the values of Component 2 were similar to Eastern European cities.

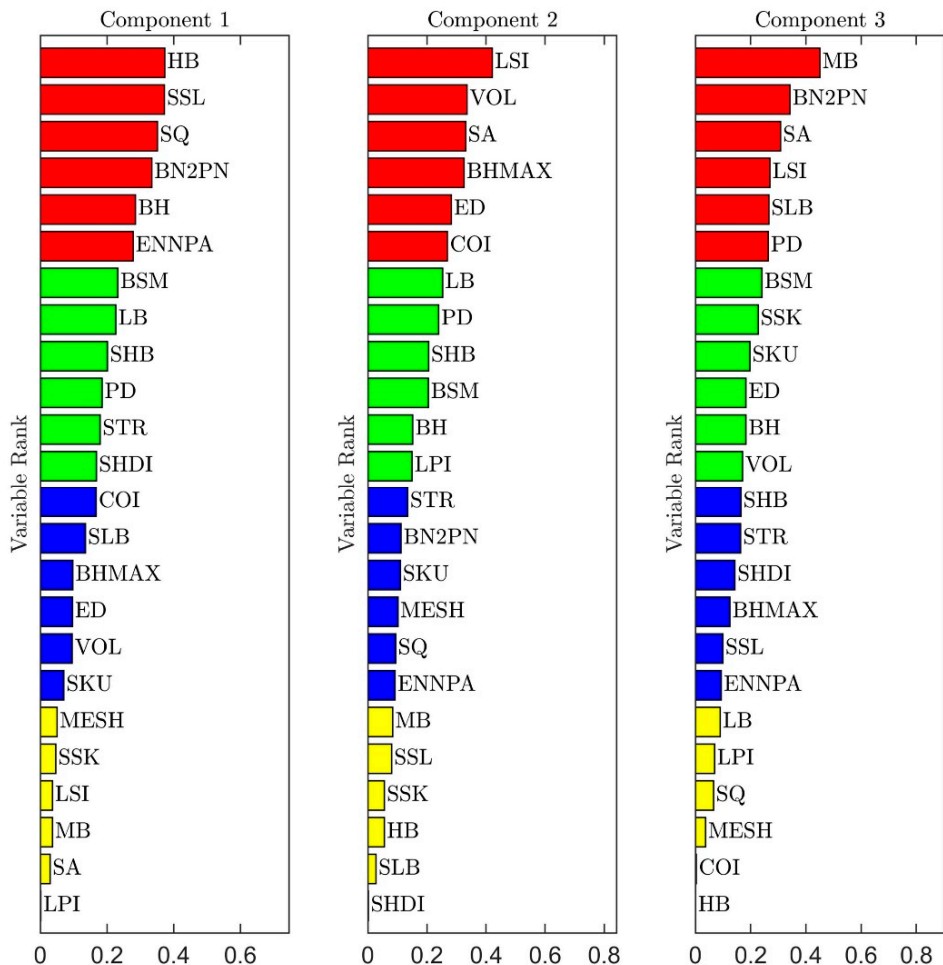

**Figure 5.** Dominant building characteristics for the first three main components. *X* axis means contribution of metrics on each component.

## 4. Discussion

In this study, a set of 3D building metrics was applied to measure and compare built landscapes in terms of their composition and configuration, avoiding the insufficiencies of traditional methods. The selected metrics can effectively distinguish the spatial heterogeneity of urban building characteristics in China and Europe. The PCA identified a very significant clustering of European and Chinese cities; however, whether the proposed metrics play a key role for this phenomenon needs additional experiments. Can the traditional metrics (e.g., building height, building density, sky view factor, and spatial proportion) do the same without these new metrics? Figure 6 displays the PCA results using only traditional buildings metrics, and the results do not reveal any significant pattern for Chinese and European cities. Compared with Figure 4, these differences demonstrate the huge potential of landscape metrics in measuring urban morphology. While traditional metrics consider the building height and spatial proportion of different building types [29], they neglect the spatial structures and texture features of building layouts. Figure 5 indicates that combining building height, building complexity, building surface fluctuations, and building spacing explains the differences in urban morphology between China and Europe. Traditional metrics only explained variations of building height; however, the landscape metrics proposed and used here can complement this with more detailed building information. This is advantageous because it reflects a more realistic building morphology and addresses the complexity and diversity of urban forms.

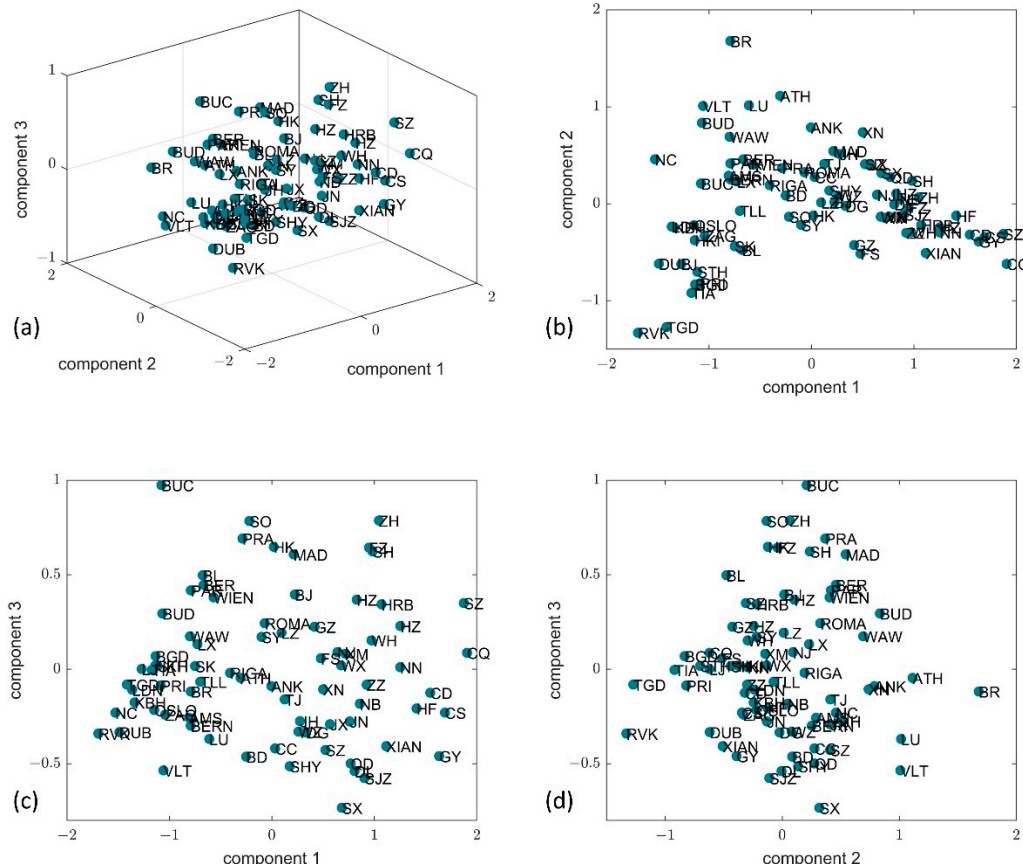

**Figure 6.** (**a**) Results of the first, second, and third principal components using only traditional building metrics; (**b**) the first component versus second component; (**c**) the first component versus third component; (**d**) the second component versus third component.

The first component in Figure 4 represents the differences in urban morphology between China and Europe. However, this component is unaffected by differences in the local genesis and urbanization processes of these cities. Of the European cities studied here, most have a long history, which means their urban development took place over a long period of time. Building heights and the proportion of high-rise buildings, which make a significant contribution to Component 1, do not differ significantly across European cities. The Chinese cities analyzed are mostly metropolises or provincial capitals. They have been modernized in recent decades and have consistently tall buildings and large building spacing; therefore, Component 1 cannot distinguish them well. However, cities vary according to their size due to urban population and economic development levels. As a result, the characteristics of total area, total volume, and building edges differ significantly between cities, even in the same region. This explains why Component 2 is sensitive to the degree of urban development and can distinguish large cities from others. Comparing the values of Component 2, some Chinese cities such as Shanghai and Guangzhou have reached a very similar level to Berlin; however, they still have significant differences to London and Paris. Another point is that most Chinese cities have similarities with the capitals of Eastern European countries. Considering that China has started modernization and urbanization since the 1980s, this study is direct evidence of the high pace of urbanization and urban transformation in China.

Despite a large body of literature dealing with urban landscape metrics, the systematic comparison of building features in 3D space, especially the spatial structure of building footprints, is not yet well documented [8,9,18,20]. Three-dimensional landscape metrics can fill this gap and contribute to a global-scale comparison of urban morphology with

more targeted and efficient indicators. With the development of high-resolution remote sensing technologies, products for building footprints and heights have been released at national, continental, and global scales [30–33]. In fast developing countries, such as China, India, and Brazil, urbanization is undoubtedly having a significant impact on urban morphology [8,9,15]—quantifying the extent of this impact is still challenging for urban researchers. Figure 4 shows a clear variability between cities and a more aggregated pattern of Chinese cities in comparison with European cities. The 3D landscape metrics can also be used to find associations with socioeconomic activities. For example, BSM and ENN metrics are designed to reflect the compactness of the spatial arrangements of buildings. An irregular and compact building layout may cause traffic congestion and increased energy consumption [8]. These findings provide meaningful support for future studies of changes in urban morphology.

However, this study fails in temporal analysis. Time is a fundamental component for urban morphology research due to continuous urban transformations and replacements [1,2]. The extent and speed of change varies from city to city due to economics and culture. Furthermore, the dynamics of urban morphology relate not only to the geometric features of buildings but also to building materials, roofing styles, street patterns, and urban green spaces. It is essential to study the complex changes in urban physical forms to improve urban planning and management.

Over the past few decades, the effects of urban morphology on the urban thermal environment have been studied [34,35]. Most of these studies applied the moving window method, aiming to propose feasible scientific suggestions for urban temperature mitigation. The metrics and software proposed in this study can complement this research and provide useful novel approaches to study the urban heat environment over multiple scales and multiple cities in a more efficient way. At different scales, the dominant building characteristics in each window may vary between 3D (building height, surface area, and volume), 2D (building coverage ratio, building number, and building spacing), and configuration features. Compared with the conclusions made at a single scale and city, multiscale studies covering different climate zones are conducive to revealing the influencing mechanisms in much more depth [8,32], allowing for researchers to make targeted proposals for local urban planning from the perspective of heat management.

## 5. Conclusions

Applying 3D landscape metrics and principal component analysis, this study demonstrated the effectiveness of selected metrics in describing the spatial heterogeneity of urban buildings, and identified a significant difference in urban morphology between China and Europe. The height, surface fluctuation, and spatial proportion of sub-high and high buildings in China significantly exceeded that of European cities, while the latter show higher building density, narrower distances among buildings, and a weaker texture direction of building layouts. Among selected cities, the urban morphology in China is more uniform than in European cities, as indicated by a smaller diameter of the circumscribing ellipse and a more aggregated pattern in the PCA results. To be specific, the first principal component explains 48.9% of variability and can well distinguish between the urban morphology in Europe and China. For this component, building height, surface fluctuation, building complexity, and nearest distances among buildings contributed more than other features. The second component is sensitive to the degree of urban development. Large metropolitan and province capitals are well separated from other cities, with less urban population and relatively slow economic development. This property is obvious among the selected cities in Europe, and has a descending order: western, eastern and northern parts. Configuration metrics, maximum building height, surface area, and volume contributed most to this component. The main reason for the differences in urban morphology between China and Europe could be rapid urbanization; however, this conclusion requires further investigations. Unlike the selected cities in China, the cities in Europe were chosen from 36 countries or regions with different architecture cultures and climate conditions. Future

studies should therefore select more cities of various sizes around the world and combine them with remote sensing earth observation products to further investigate the impact of urbanization on urban morphology over time.

**Author Contributions:** Conceptualization, F.G. and U.S.; methodology, F.G. and U.S.; software, F.G.; formal analysis, F.G., W.W. and A.M.; writing—original draft preparation, F.G.; writing—review and editing, F.G., U.S., W.W. and A.M. All authors have read and agreed to the published version of the manuscript.

**Funding:** The first author would like to express his gratitude for the research support from the China Scholarship Council under Grant No. 202008080124.

**Data Availability Statement:** For European buildings, please see https://land.copernicus.eu/local/urban-atlas/building-height-2012 (accessed on 1 July 2022). For Chinese buildings, please see https://www.resdc.cn/Default.aspx (accessed on 1 July 2022).

**Conflicts of Interest:** The authors declare no conflict of interest.

## Appendix A

**Table A1.** Selected cities in China and Europe.

| Chinese Cities | Abb. | Chinese Cities | Abb. |
|---|---|---|---|
| BAODING | BD | NANJING | NJ |
| BEIJING | BJ | NANNING | NN |
| CHANGCHUN | CC | NINGBO | NB |
| CHANGSHA | CS | QINGDAO | QD |
| CHENGDU | CD | SANYA | SY |
| CHONGQING | CQ | SHANGHAI | SH |
| DALIAN | DL | SHAOXING | SX |
| DONGGUAN | DG | SHENYANG | SHY |
| FOSHAN | FS | SHENZHEN | SZ |
| FUZHOU | FZ | SHIJIAZHUANG | SJZ |
| GUANGZHOU | GZ | SUZHOU | SUZ |
| GUIYANG | GY | TIANJIN | TJ |
| HAERBIN | HEB | WENZHOU | WZ |
| HAIKOU | HK | WUHAN | WH |
| HANGZHOU | HZ | WUXI | WX |
| HEFEI | HF | XIAMEN | XM |
| HUIZHOU | HZ | XIAN | XIAN |
| JIAXING | JX | XINING | XN |
| JINAN | JN | ZHENGZHOU | ZZ |
| JINHUA | JH | ZHUHAI | ZH |
| LANZHOU | LZ | | |
| European Cities | Abb. | European Cities | Abb. |
| RIGA | RIGA | REYKJAVIK | RVK |
| AMSTERDAM | AMS | SKOPJE | SK |
| ATHINA | ATH | SOFIA | SO |
| BERN | BERN | TALLINN | TLL |
| BRATISLAVA | BL | TIRANA | TIA |
| BRUSSEL | BR | VALLETTA | VLT |
| BUCURESTI | BUC | WARSZAWA | WAW |
| BUDAPEST | BUD | WIEN | WIEN |
| HELSINKI | HKI | ZAGREB | ZAG |
| KOBENHAVN | KBH | ROMA | ROMA |
| LEFKOSIA | NC | LONDON | LDN |
| LJUBLJANA | LJ | PARIS | PAR |
| LUXEMBOURG | LU | BERLIN | BER |
| MADRID | MAD | STOCKHOLM | STH |
| OSLO | OSLO | LISBOA | LX |
| PODGORICA | TGD | BEOGRAD | BGD |
| PRAHA | PRA | ANKARA | ANK |
| PRISTINA | PRI | DUBLIN | DUB |

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
