# Peer review of "Differences in Urban Morphology between 77 Cities in China and Europe"

_remotesensing, doi:10.3390/rs14215462_

Round 1
Reviewer 1 Report
Dear Authors,
As the call indicates, the article has to work with urban morphology - your work failed on this point. I recommend reading the recommended article by Anne Vernez Moudon, then studying the urban morphological methods (historico-geographical, typo-morphological, configurational and the new methods like Morphometrics, Morpho) before concluding, because these sentences are far from reality.
"The restricted availability of methods describing the urban morphology is another 59 limit to a large-scale comparison. Common metrics for measuring urban building characteristics mainly include building height, building density and spatial proportion of various building classes [8]. "
The urban morphological part is missing and because of that, the method you used created results if you would like to link them with the urban heat islands. I am not sure either that the manuscript is dealing adequately with the problem of urban ecology.
The methodology needs more effort not to put the European towns in one cluster and forget the high variability of the built environment in these towns.
I appreciate the data collection and the data distribution, but based on the selected towns the discussion and the conclusion should be more specific and less general.
There are a lot of 'if' and "would". The manuscript indicates that it is not complete and the final results and conclusions need more research to be accurate.
"If we could obtain building height 256 information and temporal changes for many more cities around the world, it would be 257 interesting and important to see how urban morphology varies globally with space and time." - there are studies, books and articles on this topic. Study them and please do not write down such an evident "finding".
I am looking forward to reading your updated article!
Author Response
Please see attached file.
Response to reviewers
The very helpful recommendations of the reviewers are gratefully acknowledged. In this letter we respond to each suggestion. Our comments are inserted in blue color following each point of the reviewers. The line numbers cited in our response refer to the revised manuscript with no changes marked.
Summary of major changes based on Reviewers’ comments
Based on the comments and recommendations raised by the reviewers, we have revised the manuscript significantly. Major changes include: 1) in the Introduction section, we added more accurate descriptions related to urban morphology and outlined our research focus; 2) considering that all reviewers asked for a more detailed discussion of the variabilities of urban morphology in Europe, we added one paragraph describing the differences and similarities within Europe as well as that between China and Europe; 3) we revised the title and some unsuitable expressions in the manuscript structure and figures.
Reviewer 1's comments
1) As the call indicates, the article has to work with urban morphology - your work failed on this point. I recommend reading the recommended article by Anne Vernez Moudon, then studying the urban morphological methods (historico-geographical, typo-morphological, configurational and the new methods like Morphometrics, Morpho) before concluding, because these sentences are far from reality.
"The restricted availability of methods describing the urban morphology is another limit to a large-scale comparison. Common metrics for measuring urban building characteristics mainly include building height, building density and spatial proportion of various building classes [8]. "
The urban morphological part is missing and because of that, the method you used created results if you would like to link them with the urban heat islands. I am not sure either that the manuscript is dealing adequately with the problem of urban ecology.
We thank Reviewer 1 for the useful suggestions. Both of them are involved in a more accurate and comprehensive description of urban morphology. We have checked the books and articles recommended by reviewer 1 and re-wrote the first paragraph. Please see lines 37-48. In the modified version, we summarized the fundamental components as well as the relationship with urban ecology, particularly urban heat islands. In addition, in the second paragraph (lines 49-57), we outlined that this study is focused on the definition of morphological characteristics and follows a spatial analytical approach analyzing ‘built forms’ using satellite images at high resolution of pixels.
2) The methodology needs more effort not to put the European towns in one cluster and forget the high variability of the built environment in these towns.
We thank reviewer 1 for pointing out this problem. In section 3.1, we modified the description related to the differences of building characteristics among various parts of European cities. The differences and similarities between Chinese and European cities were also considered (see lines 233-244).
3) I appreciate the data collection and the data distribution, but based on the selected towns the discussion and the conclusion should be more specific and less general.
We thank reviewer 1 for this advice. In the section Discussion, we added one paragraph to discuss some specific points related to the pattern of urban morphology in China, and in Europe (lines 267-284).
4) There are a lot of 'if' and "would". The manuscript indicates that it is not complete and the final results and conclusions need more research to be accurate.
We thank Reviewer 1 for the suggestions. We have checked our manuscript and revised unclear descriptions.
5) "If we could obtain building height information and temporal changes for many more cities around the world, it would be interesting and important to see how urban morphology varies globally with space and time." - there are studies, books and articles on this topic. Study them and please do not write down such an evident "finding".
We strongly agree with reviewer 1. We have read related books and articles and modified this paragraph (improved in lines 289-293 and also lines 300-306).

Reviewer 2 Report
The importance of such studies is given by applied methodology, and by useful information on relevant differences between two distinct parts of the “urban Eurasia”. The readers are convicted that urbanization process is about the same, even if the urban landscape is different, and that comparative morphology evaluation of cities can be made by a quantitative approach.
Comments and suggestions:
11) Regarding the paper’s Title I have a comment and a proposal:
- a discussion point looks to “significant” differences! The manuscript has a lot of elements to demonstrate that applying 3D landscape metrics and PCA are defined certain morphological indicators, which measure the differences by their significance degree. I agree this way, but I recommend, just at the beginning, a short and explicit explanation: why you consider these differences significant?
- a proposal: I believe that the current Title requires ……. selected cities).
2) The Abstract is well structured and attractive, inciting the readers.
3) In my opinion, the Introduction covers a large spectrum of literature regarding the quantification of urban morphology, and other issues connected with the field research. Using the results obtained by other scholars studying the landscape metrics, the authors found an appropriate way to develop and to apply, in an integrated manner, the main concepts and methods to highlight the differences between Europe and China cities.
Please clarify the source of the Figure 1, because inside of the text you have cited Guo et al. (2021), and nothing inside of figure explanation! This figure is your creation?
For the design of paper, I suggest moving up the Figure 1 from the end of chapter. Usually, any chapter has at the end some text phrases, and not a figure or table.
4) Materials and Methods section has a good structure, each sub-chapter is well developed, and “pigmented” with specific and relevant literature. The readers find enough elements for a better understanding of the main methodological tools and steps used by authors.
A special appreciation for the map, which gives to analyst the spatial distribution of the selected cities, representing a good basis for other correlative ideas and research approach.
Two suggestions regarding the Table 1: a) please, separate the 2D and 3D Compositions, using a doted line, because both belong to the same category; b) please, make distinction between the Table Title and other elements, which can be placed bellow of the table! At the same time revise the unclear phrase “Bold metrics are currently widely used building metrics”.
5) Results section contains the main findings regarding the landscape metrics for the three groups and the Chinese and European cities’ clusters using the building metrics. The results presentation is acceptable, but the titles of both sub-sections require some improvements: a) I suggest to modify the title of the first subsection in Different metrics of cities’ groups for China and Europe (or a similar formula); b) For the second sub-section, please, highlight the main “product” of PCA application, not the method (usually, such sub-section should reflect the concrete result, otherwise the current title repeat, in a great proportion another title from the above section)!.
6) I highly appreciate the Discussion on the results, the comments, and the Figure 6, which put together the selected cities. I believe that for a critical discussion it would have been interesting to highlight the specific urban morphology of some cities. For example, your applied methodology reflects a kind of distinction between Eastern and Western Europe cities, taking into consideration that the first category preserved enough elements coming from communist period? They (Eastern European cities) have more similarities with the Chinese cities?
7) I have no comments on Conclusion section, excepting the next suggestion. Conforming with a correct design of an article, the end of a manuscript should have a synthetic phrase/paragraph, not an enumeration, picture, or table! However, the scientific literature has numerous examples that contradict me.
Author Response
Please see attached file!
Response to reviewers
The very helpful recommendations of the reviewers are gratefully acknowledged. In this letter we respond to each suggestion. Our comments are inserted in blue color following each point of the reviewers. The line numbers cited in our response refer to the revised manuscript with no changes marked.
Summary of major changes based on Reviewers’ comments
Based on the comments and recommendations raised by the reviewers, we have revised the manuscript significantly. Major changes include: 1) in the Introduction section, we added more accurate descriptions related to urban morphology and outlined our research focus; 2) considering that all reviewers asked for a more detailed discussion of the variabilities of urban morphology in Europe, we added one paragraph describing the differences and similarities within Europe as well as that between China and Europe; 3) we revised the title and some unsuitable expressions in the manuscript structure and figures.
Reviewer 2's comments
The importance of such studies is given by applied methodology, and by useful information on relevant differences between two distinct parts of the “urban Eurasia”. The readers are convicted that urbanization process is about the same, even if the urban landscape is different, and that comparative morphology evaluation of cities can be made by a quantitative approach.
Comments and suggestions:
1) Regarding the paper’s Title I have a comment and a proposal:
- a discussion point looks to “significant” differences! The manuscript has a lot of elements to demonstrate that applying 3D landscape metrics and PCA are defined certain morphological indicators, which measure the differences by their significance degree. I agree this way, but I recommend, just at the beginning, a short and explicit explanation: why you consider these differences significant?
- a proposal: I believe that the current Title requires ……. selected cities).
We thank reviewer 2 for the suggestions on the title. We changed our title to ‘Differences of urban morphology between 77 cities in China and Europe’. The differences of urban morphology have been considered in selected cities, and we deleted the word ‘significant’ because we did not make statistical significance tests.
2) The Abstract is well structured and attractive, inciting the readers.
We thank reviewer 2 for the appreciation of the Abstract.
3) In my opinion, the Introduction covers a large spectrum of literature regarding the quantification of urban morphology, and other issues connected with the field research. Using the results obtained by other scholars studying the landscape metrics, the authors found an appropriate way to develop and to apply, in an integrated manner, the main concepts and methods to highlight the differences between Europe and China cities.
Please clarify the source of the Figure 1, because inside of the text you have cited Guo et al. (2021), and nothing inside of figure explanation! This figure is your creation?
For the design of paper, I suggest moving up the Figure 1 from the end of chapter. Usually, any chapter has at the end some text phrases, and not a figure or table.
We agree with reviewer 2 about the location of Figure 1. This figure is created by us to show the texture feature for different building layouts. In the modified version, we moved this figure into section Methods (now Fig. 2) for a better understanding of the metric ‘texture direction aspect ratio (STR)’. Also, we added one sentence to describe the source of photographs in this figure (line 158).
4) Materials and Methods section has a good structure, each sub-chapter is well developed, and “pigmented” with specific and relevant literature. The readers find enough elements for a better understanding of the main methodological tools and steps used by authors.
A special appreciation for the map, which gives to analyst the spatial distribution of the selected cities, representing a good basis for other correlative ideas and research approach.
Two suggestions regarding the Table 1: a) please, separate the 2D and 3D Compositions, using a dotted line, because both belong to the same category; b) please, make distinction between the Table Title and other elements, which can be placed bellow of the table! At the same time revise the unclear phrase “Bold metrics are currently widely used building metrics”.
We thank reviewer 2 for the appreciation and advice. In Table 1, we used a dotted line to replace the solid line as suggested. Although we did not put the table title below the table, due to the requirements of the journal’s template, we reset the paragraph spacing in the modified version. In addition, we deleted the last sentence for a clearer description, because we have mentioned this in other parts (e.g., sections Introduction and Discussion).
5) Results section contains the main findings regarding the landscape metrics for the three groups and the Chinese and European cities’ clusters using the building metrics. The results presentation is acceptable, but the titles of both sub-sections require some improvements: a) I suggest to modify the title of the first subsection in Different metrics of cities’ groups for China and Europe (or a similar formula); b) For the second sub-section, please, highlight the main “product” of PCA application, not the method (usually, such sub-section should reflect the concrete result, otherwise the current title repeat, in a great proportion another title from the above section)!.
We thank reviewer 2 for this suggestion very much. We changed the title of sub-sections. Now, the title of section 3.1 is ‘Different metrics of city groups in China and Europe’, as suggested. The title of section 3.2 is ‘More uniform urban morphology in China than in Europe’.
6) I highly appreciate the Discussion on the results, the comments, and the Figure 6, which put together the selected cities. I believe that for a critical discussion it would have been interesting to highlight the specific urban morphology of some cities. For example, your applied methodology reflects a kind of distinction between Eastern and Western Europe cities, taking into consideration that the first category preserved enough elements coming from communist period? They (Eastern European cities) have more similarities with the Chinese cities?
We thank reviewer 2 for the advice. We added one paragraph to discuss the differences and similarities of urban morphology among the selected European and Chinese cities (now lines 267-284).
7) I have no comments on Conclusion section, excepting the next suggestion. Conforming with a correct design of an article, the end of a manuscript should have a synthetic phrase/paragraph, not an enumeration, picture, or table! However, the scientific literature has numerous examples that contradict me.
We thank reviewer 2 for the suggestion to the Conclusions section. We modified the structure and style of this section accordingly. Please see lines 320-343

Reviewer 3 Report
The manuscript is generally well-written and structured. However, in my opinion, the manuscript has some shortcomings in regard to data analyses, and the scope of work. Given the following shortcomings, the manuscript requires minor revisions.
'Figure 2. Spatial locations of selected cities in China and Europe' can be transformed into a GIS-ready map. The study area map requires better visualization of cities with proper annotation.
Discuss in detail the building footprint dataset with a better description of errors. It would be better if city-specific descriptions can be provided.
City-specific representations (maps) for morphological factors and their discussions need to be incorporated into the results.
Besides PCA analysis, there is a requirement to provide a detailed clustering or classification of cities as per their common morphological characteristics.
Author Response
Please see attached file!
Response to reviewers
The very helpful recommendations of the reviewers are gratefully acknowledged. In this letter we respond to each suggestion. Our comments are inserted in blue color following each point of the reviewers. The line numbers cited in our response refer to the revised manuscript with no changes marked.
Summary of major changes based on Reviewers’ comments
Based on the comments and recommendations raised by the reviewers, we have revised the manuscript significantly. Major changes include: 1) in the Introduction section, we added more accurate descriptions related to urban morphology and outlined our research focus; 2) considering that all reviewers asked for a more detailed discussion of the variabilities of urban morphology in Europe, we added one paragraph describing the differences and similarities within Europe as well as that between China and Europe; 3) we revised the title and some unsuitable expressions in the manuscript structure and figures.
Reviewer 3's comments
The manuscript is generally well-written and structured. However, in my opinion, the manuscript has some shortcomings in regard to data analyses, and the scope of work. Given the following shortcomings, the manuscript requires minor revisions.
1) 'Figure 2. Spatial locations of selected cities in China and Europe' can be transformed into a GIS-ready map. The study area map requires better visualization of cities with proper annotation.
We thank reviewer 3 for the suggestions. We modified the manuscript and map accordingly (line 114).
2) Discuss in detail the building footprint dataset with a better description of errors. It would be better if city-specific descriptions can be provided.
We thank reviewer 3 for the suggestion on the data source. For European cities, the document describing the accuracy of building heights was downloaded together with the raster data. In the modified version, we added a description of this in section 2.2 and provided the website in line 352. For Chinese cities, we evaluated the accuracy through published literature (line 123).
3) City-specific representations (maps) for morphological factors and their discussions need to be incorporated into the results.
We thank reviewer 3 for the advice. Actually, we calculated the local building characteristics using the moving window method, which can show the spatial heterogeneity of urban morphology within a city, however, this did not fall into the topic of this study. This study is focused on a large-scale comparison among cities, therefore, we only listed the specific values for the entire urban built-up landscape. That means one value for each metric for each city. For future studies related to the urban heat environment, we will display city-specific maps of morphological metrics.
4) Besides PCA analysis, there is a requirement to provide a detailed clustering or classification of cities as per their common morphological characteristics.
We thank reviewer 3 for the suggestion. We added more detailed expressions in section Results and section Discussion (lines 230-244 and also lines 267-284).

Round 2
Reviewer 1 Report
Thank you for your improvements!